# Long-Term Kinetics of SARS-CoV-2 Neutralizing and Anti-Receptor Binding Domain Antibodies among Laboratory-Confirmed COVID-19 Cases in Delhi National Capital Region, India: A Prospective, One-Year Follow-Up Study

**DOI:** 10.3390/jcm13030762

**Published:** 2024-01-29

**Authors:** Puneet Misra, Guruprasad R. Medigeshi, Shashi Kant, Abhishek Jaiswal, Mohammad Ahmad, Anisur Rahman, Randeep Guleria, Sanjay Kumar Rai, Trideep Jyoti Deori, Suprakash Mandal, Gaurav Gongal, Mohan Bairwa, Partha Haldar, Rakesh Kumar, Neha Garg

**Affiliations:** 1Centre for Community Medicine, Old OT-Block, All India Institute of Medical Sciences, Ansari Nagar, New Delhi 110029, India; skant76@gmail.com (S.K.); drsanjay.aiims@gmail.com (S.K.R.); trideep171@gmail.com (T.J.D.); drsuprakashcm@gmail.com (S.M.); dr.ggongal@gmail.com (G.G.); mohanbairwa@aiims.edu (M.B.); drparthohaldar@gmail.com (P.H.); drrakesh@hotmail.co.uk (R.K.); 2Translational Health Science and Technology Institute, Faridabad 121001, India; gmedigeshi@thsti.res.in (G.R.M.); bioassay_qm@thsti.res.in (N.G.); 3Employee State Insurance Corporation Medical College & Hospital, Faridabad 121001, India; jaiswal.aiims@gmail.com; 4WHO Country Office, New Delhi 110011, India; ahmadmoh@who.int (M.A.); rahmanan@who.int (A.R.); 5Institute of Internal Medicine & Respiratory and Sleep Medicine, Medanta, Gurugram 122001, India; randeepguleria2002@yahoo.com

**Keywords:** SARS-CoV-2, COVID-19, antibody, PRNT, neutralizing antibody

## Abstract

**Background:** This study was conducted with the objective of measuring the neutralizing and anti-receptor binding domain antibody levels against SARS-CoV-2 among laboratory-confirmed COVID-19 cases and exploring its long-term kinetics over a period of 1 year. **Methods:** One hundred laboratory-confirmed COVID-19 cases were recruited. Serum samples of the participants were collected within three months from the date of the positive COVID-19 report. The participants were prospectively followed up every three months for symptoms and the collection of blood samples for three additional rounds. The presence of anti-SARS-CoV-2 antibodies (IgA, IgG, and IgM antibodies), anti-receptor binding domain antibodies (anti-RBD), and neutralizing antibodies were measured. **Findings:** Median plaque reduction neutralization test (PRNT) titers showed a rising trend in the first three rounds of follow-up. The quantitative anti-receptor binding domain ELISA (QRBD) values showed a declining trend in the initial three rounds. However, both the PRNT titers and QRBD values showed significantly higher values for the fourth round of follow-up. Total antibody (WANTAI) levels showed an increasing trend in the initial three rounds (statistically significant). **Interpretation:** Neutralizing antibodies showed an increasing trend. The anti-receptor binding domain antibodies showed a decreasing trend. Neutralizing antibodies and anti-RBD antibodies persisted in the majority.

## 1. Introduction

The SARS-CoV-2 virus, a single-stranded RNA virus from the genus Betacoronavirus, is responsible for Corona Virus Disease (COVID-19) [1]. It first appeared in Wuhan, China, in December 2019, and on 11 March 2020, the WHO declared it a global pandemic. From cough, fever, and malaise to severe pneumonia and acute respiratory distress syndrome, SARS-CoV-2 infection results in a wide spectrum of clinical symptoms [2,3].

Antibody-mediated (humoral immunity) immunity is thought to play a vital role in protection, both in naturally infected and vaccinated people. The SARS-CoV-2 virus induces a classic viral response in which IgM antibodies appear first, followed by IgG antibodies, which remain detectable for several months post symptom onset (PSO), while IgM declines by 2–3 weeks of PSO [4]. There are several serological techniques available to identify these antibodies, including chemiluminescent immunoassay, lateral flow immunoassay, and enzyme-linked immunosorbent assay [5]. Serological tests are crucial in epidemiological surveys because they can be used to find asymptomatic and previously undetected infections. Neutralizing antibodies, which can neutralize the virus and offer protection against subsequent infection, are of great relevance. The plaque reduction neutralization test (PRNT) is an accepted gold standard for identifying neutralizing antibodies. The human acetylcholine esterase-2 (ACE-2) receptor is contacted by the receptor-binding protein found in the virus’ spike protein (S), which facilitates the virus’ entrance into host cells [6,7,8]. The most effective neutralizing epitope affording protection against SARS-CoV-2 works by preventing the interaction between the S protein receptor-binding domain (RBD) of the virus and ACE-2 of the human host [9,10]. Understanding the antibody kinetics of Anti-SARS-CoV-2 antibodies is pivotal in the battle against COVID-19. There have been studies conducted in Iceland and the United States that indicate that the antibodies can last for up to 4 months after infection. However, other research has found that the antibodies rapidly decline within 3–4 months [11]. The majority of the research that is now available has examined antibody results at 6 months. Yet, to grasp the characteristics or the patterns of antibody depletion, it is necessary to comprehend the long-term durability of SARS-CoV-2-specific IgG and IgM and neutralizing antibody responses following symptom onset or laboratory confirmation.

The present study was conducted with the objective of measuring the neutralizing and anti-receptor binding domain antibody levels against SARS-CoV-2 among laboratory-confirmed COVID-19 cases and exploring its long-term kinetics over a 1-year period.

## 2. Materials and Methods

The present study was conducted using 100 participants who were enrolled from 15 March to 31 May 2021 from two sites: one rural site at Ballabgarh, Haryana, and another urban site at Dakshinpuri, New Delhi. Participation was voluntary. All of the participants were recruited within 3 months of a positive rapid antigen test (RAT)/real-time polymerase chain reaction (RT-PCR) report for COVID-19. The participants were recruited into the present study regardless of their age or current COVID-19 disease status. The participants who refused to give written informed consent or had contraindications related to venipuncture were excluded from the study. We gathered data on the participants’ basic demographics, exposure history to COVID-19 cases, symptoms suggestive of COVID-19 in the previous three months, and clinical history from the participants who gave their consent.

Blood collection: Trained phlebotomists collected 5 mL of venous blood in plain vials from each participant within three months of testing positive for COVID-19. The serum was separated after centrifugation.

According to the prescribed optical density (OD) cut-off value, an ELISA-based assay (WANTAI^®^) was used to detect IgG antibodies specific to SARS-CoV-2. A plaque reduction neutralization test (PRNT) was used to assess the neutralizing antibodies against SARS-CoV-2. Using quantitative RBD ELISA, anti-receptor binding domain (RBD) antibody (IgG) levels were determined.

The participants were followed up for symptoms, COVID-19 vaccination status, hospitalization status, and repeated blood sample collection every 3 months, totaling four rounds of follow-up. The study was conducted from 15 March 2021 to 30 April 2022.

The first round of data collection took place from March 2021 to May 2021, the second round from June to August 2021, the third round from September to November 2021, and the fourth round from December 2021 to April 2022.

The blood samples were tested for antibodies using PRNT, QRBD, and WANTAI during the various rounds of the data collection.

Plaque reduction neutralization test (PRNT): PRNT for SARS-CoV-2 on Vero E6 cells was carried out to measure the neutralizing antibodies. The SARS-CoV-2 virus strain (B.6) with GenBank accession ID: MW422884.1 was used for the PRNT. A PRNT_50_ titer of 20 or more was classified as positive, and a PRNT50 titer of 20 or less was classified as negative.

Quantitative anti-receptor binding domain ELISA (QRBD): Quantitative enzyme-linked immunosorbent assay (ELISA) was used to estimate serum IgG antibodies binding to the receptor-binding domain of the SARS-CoV-2 Spike protein. The test reported the anti-RBD IgG antibodies in ELU/mL. QRBD ≥ 12.0 ELU/mL was reported as positive, and between 8.0 and <12.0 ELU/mL was reported as equivocal. QRBD < 8.0 ELU/mL was reported as negative. The QRBD ELISA assay was also calibrated with WHO International Standards. The NIBSC 20/136 reference standards of the WHO were used to transform the concentration values from ELU/mL to BAU/mL by generating a conversion factor. The conversion factor obtained was 1.0.

WANTAI SARS-CoV-2 Antibody ELISA: It was an enzyme-linked immunosorbent assay (ELISA) for the qualitative detection of total antibodies to SARS-CoV-2 virus in human serum or plasma specimens (anti-SARS-CoV-2 IgA, IgG and IgM antibodies). The kit is intended for the screening of patients suspected of infection with the SARS-CoV-2 virus and as an aid in the diagnosis of the coronavirus disease 2019 (COVID-19). Specimens with OD ≥ 0.19 were considered positive, and <0.19 was considered negative.

The detailed methodologies of PRNT, QRBD, and WANTAI have been published previously [12].

Statistical analysis: Categorical variables are reported as frequency and percentage. The normality of continuous variables was tested using the Shapiro–Wilk test. Continuous variables are reported as median with interquartile range. The Wilcoxon rank-sum test was applied to test the statistical significance of continuous variables.

Ethics: Ethical permission was granted by the institute ethics committee of All India Institute of Medical Sciences, New Delhi. (Ref. No.IEC-959/04.09.2020)

Role of funding source: The World Health Organization (WHO) provided financial and technical support for this study.

## 3. Results

A total of 100 laboratory-confirmed COVID-19 cases were recruited in the study and followed every three months for a total of four rounds of data collection. The paired sample could be collected for 98 participants in round 2, 91 participants in round 3, and 79 participants in round 4. The age of the participants ranged from 14 to 72 years, with a mean (SD) of 37·0 (13·5) years, the majority being males (64%). The symptomatic and vaccination profiles of the participants during the four rounds are shown in Table 1.

Most of the participants (63%) at the time of recruitment had a history of fever (63%), followed by cough (42%), sore throat (35%), and loss of taste sensation (24%). Seventy-four participants had at least one symptom, and the remaining twenty-six were asymptomatic. In the second round, 16 (16.3%) had a fever; in the third round, 16 (17.6%) had a fever; and in the fourth round, 13 (16.5%) had a fever. The average duration of fever symptoms was 3.7 days during round 1, 3.8 days during round 2, 7.2 days during round 3, and 4 days during round 4. Cough was present among 13 (13.3%), 12 (13.2%), and 11 (13.9%) participants during the 2nd, 3rd, and 4th rounds, respectively. History of seeking medical treatment was present in 13 (13%), 2 (2%), 7 (7.7%), and 0 (0%) participants in round 1, round 2, round 3, and round 4, respectively. History of hospitalization was present in 10 (10%), 2 (2.0%), 5 (5.5%), and 0 (0%) of the participants during rounds 1, 2, 3, and 4, respectively. In round 1, as only laboratory-confirmed COVID-19 cases were enrolled, all (100, 100%) had COVID-19 positive reports. During rounds 2, 3, and 4, 15 (15.3%), 0 (0%), and 1 (1.3%) of the participants had COVID-19-positive reports.

The proportion of COVID-19-vaccinated individuals increased from 22.0% during round 1, to 62.2% in round 2, 72.5% during round 3, and 93.7% during round 4.

Sixty-nine participants out of one hundred (69.0%) had neutralizing antibodies (PRNT_50_ titer ≥ 20) during round 1, which increased to seventy-two out of ninety-eight (73.5%) during round 2, seventy-two out of ninety-one (79.1%) during round 3, and finally, to seventy-six out of seventy-nine (96.2%) during round 4. The median neutralizing antibody (PRNT_50_) titer showed an increasing trend from round 1 to round 4, with the difference being statistically significant (reference being round 1) (Table 2).

Almost all of the participants (97 out of 100, 97.0%) had anti-RBD antibodies (≥12·0 ELU/mL) during round 1, which decreased to 90 out of 96 (93.8%) during round 2, 85 out of 91 (93.4%) during round 3, and 78 out of 79 (98.7%) during round 4. However, the median anti-RBD antibody levels (ELU/mL) (BAU/mL) showed a decreasing trend from round 1 to round 3 (statistically significant) and a sudden increase in the fourth round (statistically significant) (Table 2).

The total anti-SARS-CoV-2 antibodies (optical density ≥ 0.19) remained almost similar throughout the three rounds (99 out of 100 were positive in round 1, 96 out of 97 in round 2, and 91 out of 91 in round 3). The total antibody against SARS-CoV-2 (WANTAI optical density) showed an increasing trend from round 1 to round 3 (statistically significant) (Table 2).

The PRNT_50_ titer, QRBD levels and WANTAI optical density showed a non-normal distribution during all rounds of data collection (Shapiro–Wilk *p*-value < 0.0001)

Figure 1 depicts the boxplot of the natural log of PRNT titers in four rounds. There was an increasing trend in PRNT titers in the first three rounds. The increase was significantly higher in round 4. Figure 2 shows the boxplot of the natural log of QRBD levels in four rounds. It depicts a decreasing trend in QRBD levels in the first three rounds. However, there is a significant increase in round 4.

Being a health worker was associated with a higher median PRNT_50_ titer compared to those who were not health workers. However, this association was statistically significant in round 1 and round 2 only. Urban residents had higher median PRNT_50_ titer compared to those who were rural residents. This difference was statistically significant in round 3 and round 4. The participants who were hospitalized during the respective round of data collection had a higher median PRNT_50_ titer compared to those who were not hospitalized during the same round. However, this association was not statistically significant in any round of data collection. The median PRNT_50_ titer was not associated with the presence or absence of COVID-19-related symptoms. The COVID-19 vaccination status was significantly associated with the PRNT_50_ titer for the first three rounds, with vaccinated individuals having significantly higher titers compared to unvaccinated individuals. The PRNT_50_ titer of vaccinated individuals was also higher compared to unvaccinated individuals in round 4; however, the difference was not significant in round 4 (Table 3).

Urban residents had higher median QRBD levels compared to rural residents. The difference was only statistically significant in round 3. Healthcare workers had higher median QRBD levels compared to those who were not health workers. The difference was only statistically significant in round 1. Participants who were hospitalized during their respective rounds of data collection had higher median QRBD levels compared to those who were not hospitalized. However, the difference was statistically not significant in any round of data collection. The median QRBD levels were not associated with the presence or absence of COVID-19-related symptoms. COVID-19 vaccination status was significantly associated with the QRBD levels for the first three rounds, with vaccinated individuals having significantly higher levels compared to unvaccinated individuals. The QRBD levels were higher among vaccinated individuals compared to unvaccinated individuals in round 4; however, the difference was not significant in round 4 (Table 4).

## 4. Discussion

The present study was conducted using one hundred laboratory-confirmed COVID-19 cases. Every participant underwent testing for COVID-19-neutralizing antibodies, total antibodies (IgA, IgG, and IgM), and anti-RBD IgG antibodies via PRNT, WANTAI, and QRBD, respectively.

In the present study, the neutralizing antibody titer showed an increasing trend, while the anti-RBD antibodies showed a decreasing trend in the first three rounds of follow-up. However, in the fourth round, both neutralizing antibodies and anti-RBD antibodies increased. This could be because of the Omicron wave, which was ongoing during the data collection of round 4 (December 2021-April 2022) (Appendix A) [13,14]. The neutralizing antibody titer persisted in 76 cases out of all the participants included in the study. This could be because of the likely repeated exposure to the COVID-19 virus, which was inevitable due to the 2nd and 3rd waves of COVID-19 cases in India. Additionally, during the first round, only 22% of the participants were vaccinated, but by the fourth round, almost all (94%) of the participants were vaccinated.

Our findings on the persistence of neutralizing antibodies is in agreement with previous studies [15,16,17]. Neutralizing antibodies have been reported to persist for at least 6 to 12 months after infection [18,19,20,21,22,23,24]

However, the antibodies against the receptor-binding domain (QRBD) levels showed a declining trend till round 3, though it also persisted. The persistence of antibodies against spike proteins has also been found in previous studies [15,16].

The total anti-SARS-CoV-2 antibodies also remained positive for almost all the participants throughout round 3. We could not test the round 4 samples because of the unavailability of WANTAI test kits.

The PRNT titer was higher among individuals who worked as health professionals compared to those who did not. This could be because of repeated exposure to the higher viral load of COVID-19 among health professionals.

Those who were residents of urban areas also had higher PRNT titers compared to those who were living in rural areas. This could be because of higher population density and overcrowding, and hence, repeated exposure to COVID-19 cases.

Participants who had been hospitalized had higher PRNT titers compared to those who were not hospitalized. This can be because of higher immune response among the severe cases that required hospitalization (a proxy indicator of severity). Haveri A et al. also reported similar findings of higher neutralizing antibody titers among those who had severe infection [17].

A similar finding was also found in the QRBD levels, namely, higher levels in urban residents compared to rural, health workers compared to others, and those who were hospitalized compared to those who were not.

Previous studies have shown that the presence of antibodies to SARS-CoV-2 was associated with a significantly reduced risk of SARS-CoV-2 reinfection among healthcare workers for up to 7 months after infection [25,26]. We observed that neutralizing antibodies, anti-RBD antibodies, and total antibodies against SARS-CoV-2 persisted for at least a year after the infection in most individuals. This strongly suggests that the protection against reinfection is long-lived.

The participants who were vaccinated against SARS-CoV-2 showed higher values of neutralizing and anti-RBD antibodies during all rounds of the study, with the difference being statistically significant for only the first three rounds.

There is a major research effort to produce effective SARS-CoV-2 vaccines. However, little is known about whether immunity lasts for a long time after receiving a vaccination. Evidence from the convalescent sera of individuals who have recovered from infection may help determine the duration of persistent immunity and whether antibodies might protect against reinfection. Previous data show that, when measured as IgG antibodies against S protein or RBD and neutralizing antibodies, immune response after two doses of the SARS-CoV-2 vaccine is similar to that observed in convalescent sera from COVID-19 patients [27,28,29,30]. Evidence of the persistence of immunity after infection will help in predicting the persistence of immunity after SARS-CoV-2 vaccination.

Strengths: Only instances of COVID-19 with lab confirmation were included in the research. We measured neutralizing antibodies through the use of an PRNT assay, which is considered the gold standard. Additionally, total and anti-RBD antibodies were examined in all sera samples. All the assays used standard equipment and a standard procedure. All the sera samples were collected within three months following positive RT-PCR/RAT testing.

Limitations: Symptoms and history of contact were self-reported, making this study vulnerable to recall error.

## 5. Conclusions

This study shows the persistence of neutralizing and anti-RBD antibodies even after one year. The neutralizing antibodies also showed a statistically significant increasing trend during the 1st year; however, the same was not shown by the anti-receptor binding domain antibodies, which showed a decreasing trend with time. COVID-19-vaccinated individuals had higher levels of neutralizing and anti-RBD antibodies throughout the year.

## Figures and Tables

**Figure 1 jcm-13-00762-f001:**
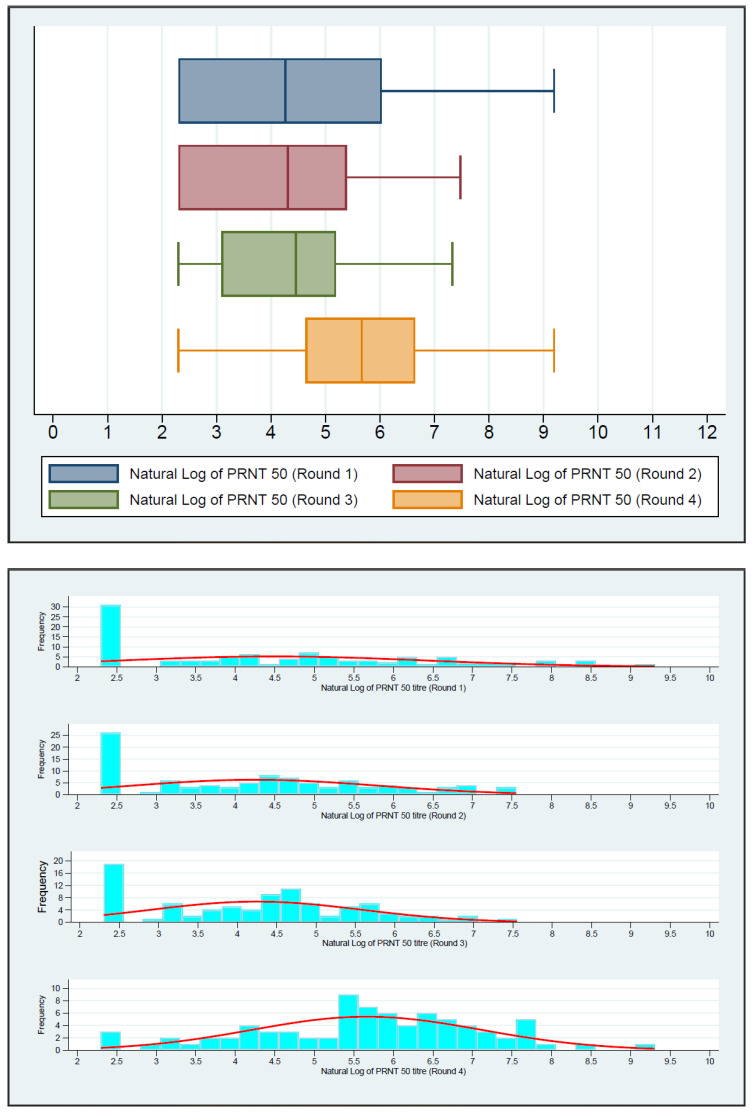
Boxplot and histogram of the trend of PRNT titers in the four rounds (Natural log of PRNT titers) showing the increasing trend of neutralizing antibodies from the round 1 to round 4.

**Figure 2 jcm-13-00762-f002:**
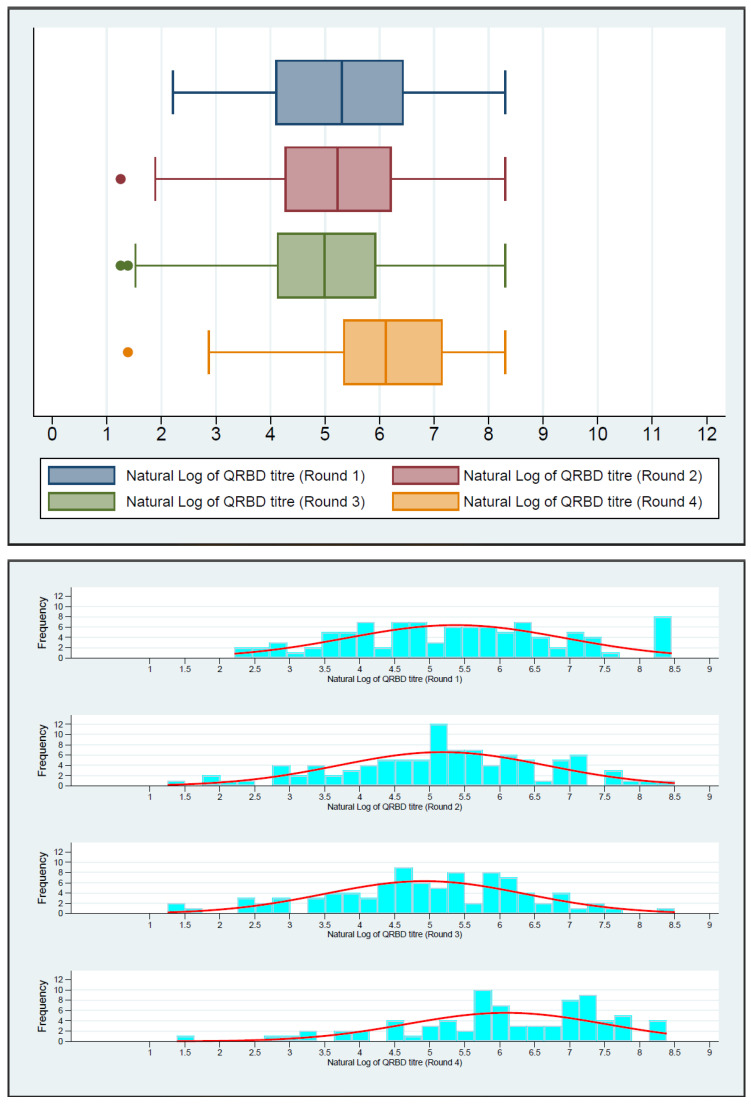
Boxplot and histogram of the trend of QRBD values in the four rounds (Natural log of QRBD values), depicting the initial fall of the median QRBD values during round 1 to 3 and then rise during round 4 to median value above round 1.

**Table 1 jcm-13-00762-t001:** Distribution of participants by selected sociodemographic and clinico-symptomatic variables.

Variables	Round 1 *n*= 100 Frequency (%)	Round 2 *n* = 98 Frequency (%)	Round 3 *n* = 91 Frequency (%)	Round 4 *n* = 79 Frequency (%)
Residence Rural	83 (83.0)	82 (83.7)	75 (82.4)	68 (86.1)
Health care worker	9 (9.0)	8 (8.2)	8 (8.8)	5 (6.3)
Fever	63 (63.0)	16 (16.3)	16 (17.6)	13 (16.5)
Sore throat	35 (35.0)	10 (10.2)	9 (9.9)	8 (10.1)
Cough	42 (42.0)	13 (13.3)	12 (13.2)	11 (13.9)
Loss of taste	24 (24.0)	0 (0.0)	2 (2.2)	0 (0.0)
Running nose	18 (18.0)	9 (9.2)	10 (11.0)	6 (7.6)
Loss of appetite	17 (17.0)	0 (0.0)	1 (1.1)	0 (0.0)
Loss of smell	16 (16.0)	1 (1.0)	1 (1.1)	0 (0.0)
Shortness of breath	13 (13.0)	0 (0.0)	0 (0.0)	0 (0.0)
Fatigue	12 (12.0)	0 (0.0)	0 (0.0)	0 (0.0)
Muscle ache	12 (12.0)	3 (3.1)	0 (0.0)	0 (0.0)
Headache	11 (11.0)	1 (1.0)	0 (0.0)	4 (5.1)
Joint ache	10 (10.0)	2 (2.0)	1 (1.1)	0 (0.0)
Nausea	9 (9.0)	1 (1.0)	0 (0.0)	0 (0.0)
Chills	8 (8.0)	0 (0.0)	2 (2.2)	1 (1.3)
Vomiting	7 (7.0)	3 (3.1)	1 (1.1)	3 (3.8)
Diarrhoea	4 (4.0)	3 (3.1)	1 (1.1)	0 (0.0)
Conjunctivitis	3 (3.0)	0 (0.0)	1 (1.1)	0 (0.0)
Rash	2 (2.0)	0 (0.0)	1 (1.1)	0 (0.0)
Nose bleeding	2 (2.0)	0 (0.0)	0 (0.0)	0 (0.0)
Altered consciousness	2 (2.0)	1 (1.0)	0 (0.0)	0 (0.0)
Seizure	1 (1.0)	0 (0.0)	0 (0.0)	0 (0.0)
Others	1 (1.0)	0 (0.0)	0 (0.0)	0 (0.0)
Any symptoms	70 (70.0)	31 (31.6)	23 (25.3)	18 (22.8)
Sought medical treatment	13 (13.0)	2 (2.0)	7 (7.7)	0 (0.0)
Missed duty because of illness	15 (15.0)	2 (2.0)	1 (1.1)	1 (1.3)
Hospitalized	10 (10.0)	2 (2.0)	5 (5.5)	0 (0.0)
History of contact with COVID-19 case	38 (38.0)	0 (0.0)	0 (0.0)	0 (0.0)
COVID-19 vaccinated	22 (22.0)	61 (62.2)	66 (72.5)	74 (93.7)
COVID-19 testing	100 (100)	17 (17.4)	2 (2.2)	5 (6.3)
COVID-19 (+) report	100 (100)	15 (15.3)	0 (0.0)	1 (1.3)
RTPCR (+)	76 (76)	15 (15.3)	0 (0.0)	1 (1.3)
RAT (+)	24 (24)	-	-	-

**Table 2 jcm-13-00762-t002:** Distribution of median (IQR) PRNT_50_ titers, Anti-RBD antibodies (ELU/mL) (BAU/mL), and total antibodies (WANTAI, OD) with respect to various rounds of the study.

Test	Round of Follow-Up	*n*	Median (IQR)	*p*-Value(Friedman)	*p*-Value (Signed Rank)
PRNT_50_ titer	Round 1	100	71.00 (10.00–415.50)	<0.0001	Ref.
	Round 2	98	74.50 (10.00–219.00)		0.0061
	Round 3	91	86.00 (22.00–180.00)		0.0064
	Round 4	79	289.00 (103.00–770.00)		0.0229
QRBD	Round 1	100	201.97 (60.00–627.55)	<0.0001	Ref.
	Round 2	98	186.50 (70.96–503.80)		0.0147
	Round 3	91	147.20 (61.50–378.80)		0.0016
	Round 4	79	453.40 (206.80–1279.50)		0.0089
WANTAI	Round 1	100	3.45 (3.27–3.56)	0.1448	Ref.
	Round 2	97	3.47 (3.41–3.54)		0.0052
	Round 3	91	3.54 (3.50–3.63)		<0.0001

**Table 3 jcm-13-00762-t003:** Distribution of median (IQR) PRNT_50_ titers by select variables and round of follow-up.

	Round 1		Round 2		Round 3		Round 4	
	Frequency (%)	PRNT (Median (IQR))	Frequency (%)	PRNT (Median (IQR))	Frequency (%)	PRNT (Median (IQR))	Frequency (%)	PRNT (Median (IQR))
Rural	83 (83.0)	67 (10–274)	82 (83.7)	63 (10–202)	75 (82.4)	75 (21–134)	68 (86.1)	272.5 (86–591)
Urban	17 (17.0)	218 (34–861)	16 (17.6)	103 (52.5–810)	16 (17.6)	114.5 (62.5–372)	11 (13.9)	796 (546–1695)
*p*-value		0.0926		0.1301		0.0326		0.0065
Male	64 (64.0)	108 (10–497.5)	62 (63.3)	69.5 (26–241)	56 (61.5)	73.5 (24–216.5)	49 (62.0)	271 (91–770)
Female	36 (36.0)	54.5 (10–238)	36 (36.7)	75 (10–195)	35 (38.5)	95 (22–129)	30 (38.0)	359.5 (189–608)
*p*-value		0.2179		0.5592		0.9282		0.3686
<40 years	63 (63.0)	45 (10–218)	62 (63.3)	61.5 (10–241)	56 (61.5)	76 (26–132)	47 (59.5)	342 (91–663)
≥40 years	37 (37.0)	156 (45–770)	36 (36.7)	79 (17–207.5)	35 (38.5)	90 (10–204)	32 (40.5)	284 (145–1006)
*p*-value		0.0023		0.8817		0.7963		0.8417
Health care worker (Yes)	9 (9.0)	861 (410–2922)	8 (8.2)	114 (79.5–1081.5)	8 (8.8)	114.5 (62.5–393.5)	5 (6.3)	796 (546–1695)
Health care worker (No)	91 (91.0)	58 (10–236)	90 (91.8)	63 (10–215)	83 (91.2)	80 (22–167)	74 (93.7)	284 (100–742)
*p*-value		0.0003		0.0361		0.1319		0.1310
Hospitalized (+)	90 (90.0)	286 (10–770)	96 (98.0)	215 (74–791)	86 (94.5)	106 (90–272)	-	-
Hospitalized (−)	10 (10.0)	68.5 (10–279)	2 (2.0)	64 (10–200)	5 (5.5)	76 (22–167)	79 (100.0)	289 (103–770)
*p*-value		0.3568		0.6756		0.7003		-
Any Symptoms (+)	70 (70.0)	71 (10–303)	31 (31.6)	62 (10–241)	23 (25.3)	99 (25–272)	18 (22.8)	347 (173–858)
Any Symptoms (−)	30 (30.0)	76 (10–494)	67 (68.4)	75 (24–219)	68 (74.7)	76 (22–141.5)	61 (77.2)	289 (103–749)
*p*-value		0.7804		0.6027		0.3020		0.8123
COVID-19 Vaccinated (+)	22 (22.0)	590.0 (115.0–1204.0)	60 (61.2)	89.0 (28.5–295.0)	66 (72.5)	95.5 (44.0–257.0)	74 (93.7)	296.5 (135.0–749.0)
COVID-19 Vaccinated (−)	78 (70.0)	45.0 (10.0–197.0)	38 (38.8)	35.5 (10.0–107.0)	25 (27.5)	31.0 (10.0–97.0)	5 (6.3)	91.0 (43.0–1857.0)
*p*-value		0.0001		0.0112		0.0043		0.6599

**Table 4 jcm-13-00762-t004:** Distribution of median (IQR) QRBD (ELU/mL) by selected sociodemographic and clinical variables and the rounds of follow-up.

	Round 1		Round 2		Round 3		Round 4	
	Frequency (%)	QRBD (Median (IQR))	Frequency (%)	QRBD (Median (IQR))	Frequency (%)	QRBD (Median (IQR))	Frequency (%)	QRBD (Median (IQR))
Rural	83 (83.0)	199.4 (55.4–612.1)	82 (83.7)	181.9 (64.8–487.9)	75 (82.4)	128.8 (46.6–357)	68 (86.1)	385.1 (199.8–1275.5)
Urban	17 (17.0)	434 (141.9–1183.4)	16 (17.6)	222.3 (122.4–714.3)	16 (17.6)	211.9 (117.3–899.7)	11 (13.9)	1005.8 (206.8–1690.3)
*p*-value		0.0742		0.3817		0.0297		0.2602
Male	64 (64.0)	235.9 (53.8–664.2)	62 (63.3)	218.6 (84.1–487.9)	56 (61.5)	150.8 (62.4–377.6)	49 (62.0)	384 (181–1208.3)
Female	36 (36.0)	135.6 (74.3–562.4)	36 (36.7)	177.4 (45.6–518.6)	35 (38.5)	147.2 (50.1–430.2)	30 (38.0)	605.9 (311.7–1303.8)
*p*-value		0.5709		0.7795		0.8162		0.2332
<40 years	63 (63.0)	129.5 (48.7–421.7)	62 (63.3)	175.7 (64.8–521.1)	56 (61.5)	151.6 (46.1–357.4)	47 (59.5)	375.5 (181.4–1272.8)
≥40 years	37 (37.0)	414.8 (141.9–990.3)	36 (36.7)	218.6 (77.5–431.2)	35 (38.5)	146.7 (67.5– 430.2)	32 (40.5)	676.2 (263.7–1545.9)
*p*-value		0.0042		0.8828		0.7784		0.3377
Health care worker (Yes)	9 (9.0)	798.4 (441.3–1415.9)	8 (8.2)	296.2 (187.6–639.4)	8 (8.8)	212.4 (136.4–749.8)	5 (6.3)	1204.9 (589.6–1976.3)
Health care worker (No)	91 (91.0)	184.7 (55.6–560.7)	90 (91.8)	178.3 (64.8–503.8)	83 (91.2)	140.6 (50.1–376.5)	74 (93.7)	405.1 (206.8–1278.3)
*p*-value		0.0045		0.1466		0.1194		0.2745
Hospitalized (+)	90 (90.0)	566 (126.1–1183.4)	96 (98.0)	236 (26.1–445.9)	86 (94.5)	231.9 (50.1–357)	-	-
Hospitalized (−)	10 (10.0)	195.2 (59.9–589.8)	2 (2.0)	186.5 (75.9–506.2)	5 (5.5)	146.9 (63.3–378.8)	79 (100.0)	453.4 (206.8–1279.5)
*p*-value		0.2082		0.7242		0.8191		-
Any Symptoms (+)	70 (70.0)	235.9 (55.6–612.1)	31 (31.6)	180.5 (47.8–503.8)	23 (25.3)	156 (50.1–452.5)	18 (22.8)	830.4 (300.7–1976.3)
Any Symptoms (−)	30 (30.0)	180.1 (63.0–692.6)	67 (68.4)	189.6 (84.1–508.7)	68 (74.7)	138 (62.4–375.1)	61 (77.2)	424.6 (181.4–1272.8)
*p*-value		0.7152		0.9878		0.5837		0.3775
COVID-19 Vaccinated (+)	22 (22.0)	718.1 (441.3–1415.9)	60 (61.2)	284.1 (122.8–859.8)	66 (72.5)	197.5 (95.2–498.9)	74 (93.7)	488.0 (218.3–1278.3)
COVID-19 Vaccinated (−)	78 (70.0)	131.0 (52.3–372.3)	38 (38.8)	90.4 (29.3–180.5)	25 (27.5)	67.5 (26.2–140.6)	5 (6.3)	363.9 (46.4–1322)
*p*-value		<0.0001		0.0001		0.0004		0.6168

## Data Availability

Deidentified patient data will be made available upon legitimate request to the Principal Investigator after a signed data access agreement.

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
