# Peer review of "Long-Term Kinetics of SARS-CoV-2 Neutralizing and Anti-Receptor Binding Domain Antibodies among Laboratory-Confirmed COVID-19 Cases in Delhi National Capital Region, India: A Prospective, One-Year Follow-Up Study"

_jcm, 2024, doi:10.3390/jcm13030762_

Round 1
Reviewer 1 Report
Comments and Suggestions for Authors
The study carried out by Misra et al is a longitudinal investigation that involved measuring the levels of neutralizing antibodies against SARS-CoV-2, anti-receptor binding domain antibodies (anti-RBD), and anti-SARS-CoV-2 total antibodies (IgA, IgG, and IgM) in laboratory-confirmed COVID-19 cases over a one-year period. Four sample collections were made spaced three months apart. While the study produced some intriguing findings, it also highlights limitations in the work.
Some of the observations are listed below:
1. The sample size used in the study is quite small and it is further accentuated with drop outs at later time points with only 79 volunteers completing the study. This is further exemplified as the authors show that there is an increasing trend in neutralizing antibodies in sera samples collected in first three visits though there was a decrease in QRBD levels in similar visits, although on closer examination one notes that the values were in the similar range and could be biased due to various sub-categories as per socio-demographic and clinical symptoms distribution.
2. The vaccination status of the volunteers during various visits and change in their antibody profile should be provided as the roll out of vaccination drive started in 2021 in the country.
3. The sudden rise in neutralizing antibodies and total antibodies during sera samples collected during last visit may be due to vaccination or sub-clinical infection. The second wave of COVID infection associated with sample collection period led to very high number of infections in India and re-infection status of all the volunteers need to be shown in the study. It would have been good to have neutralizing antibody data against all Wuhan, Delta and Omicron variants.
4. Table 1 shows that almost 16% of volunteers has fever in last three visits and needs to be elaborated.
5. Other comments on the manuscript are:
a. It would have been good to also monitor cytokine profile of the volunteers
b. Details about the virus strain used in PRNT studies should be provided in the manuscript.
c. QRBD values are presented as ELU/mL and it would be good to report the same in BAU/mL and calibrated with international reference standard.
d. Authors mention that they could not analyse the samples received in last visit for total antibody levels as the commercial kit was unavailable. It is not clear if the kit was discontinued and why no other kit could have been used for analysis of all the samples?
e. Were the samples analysed at different collection time points or together at the end of the study to avoid inter-assay variability?
f. Figure 1 has been pasted over the text making it not possible to read it.
g. The legends for Figures 1 and 2 need to be more descriptive.
6. The conclusion section is not written well and same findings are repeated in multiple sections.
Comments on the Quality of English LanguageNo major issues noted.
Author Response
Dear Sir/Ma'am,
Please see the attachment.

Reviewer 2 Report
Comments and Suggestions for Authors
This article: Long term kinetics of SARS-CoV2 neutralizing and anti-receptor
binding domain antibodies among laboratory confirmed COVID-19 cases in
Delhi National Capital Region, India.
prospective, one year follow up study. The present study was conducted with the
objective to measure 21 the neutralizing and anti-receptor binding domain antibody
levels against SARS-CoV-2 22 among laboratory confirmed COVID-19 cases and
explore its long-term kinetics over a 23 period of 1 year.
Even though including recent and relevant literature, could be structured more clearly
and follow a line of thought.
Comments:
In the material methods:
1. In the epidemiological data what categories were included?
2. How the blood sample were tested for antibody levels of IgG, IgM, RBD-IgG,
S-IgG? chemiluminescence enzyme immunoassay.?
In the result section:
1. Among the total individuals who participated, how many round did follow-up?
Mention it.
In the discussion section
1. Vaccination information should also collect from these participants to
understand the effect of vaccination on antibody levels. Just mention it.
2. Your study found that IgG positive rates remained high and that vaccinated
individuals had higher levels of IgG antibody compared to unvaccinated
individuals. You should explain in your data.
3. Your result can be drawn for IgG subtypes and neutralizing antibodies.
Because these findings fill a gap in the kinetics of the long-term immune
response to SARS-CoV-2 and highlight the need for vaccination of the
convalescent population.
Author Response

(The authors gave the same response as above.)
